# No evidence that carotenoid pigments boost either immune or antioxidant defenses in a songbird

Rebecca E. Koch[1,2], Andreas N. Kavazis[3], Dennis Hasselquist[4], Wendy R. Hood[1], Yufeng Zhang[1,5], Matthew B. Toomey [6] & Geoffrey E. Hill[1]

Dietary carotenoids have been proposed to boost immune system and antioxidant functions in vertebrate animals, but studies aimed at testing these physiological functions of carotenoids have often failed to find support. Here we subject yellow canaries (*Serinus canaria*), which possess high levels of carotenoids in their tissue, and white recessive canaries, which possess a knockdown mutation that results in very low levels of tissue carotenoids, to oxidative and pathogen challenges. Across diverse measures of physiological performance, we detect no differences between carotenoid-rich yellow and carotenoid-deficient white canaries. These results add further challenge to the assumption that carotenoids are directly involved in supporting physiological function in vertebrate animals. While some dietary carotenoids provide indirect benefits as retinoid precursors, our observations suggest that carotenoids themselves may play little to no direct role in key physiological processes in birds.

[1] Department of Biological Sciences, Auburn University, 331 Funchess Hall, Auburn, AL 36849, USA. [2] School of Biological Science, Monash University, Room 442 Building 17, Clayton, VIC 3168, Australia. [3] School of Kinesiology, Auburn University, 301 Wire Road, Auburn, AL 36840, USA. [4] Department of Biology, Lund University, Ekologihuset, Sölvegatan 37, SE-323 62 Lund, Sweden. [5] Buck Institute for Research on Aging, 8001 Redwood Boulevard, Novato, CA 94945, USA. [6] Department of Pathology and Immunology, Washington University School of Medicine, St. Louis, MO 63110, USA. Correspondence and requests for materials should be addressed to R.E.K. (email: rebecca.adrian@monash.edu)

In both the nutraceutical and evolutionary ecology literature, dietary carotenoids are widely proposed to boost immune system and antioxidant functions in vertebrate animals[1–5]. Data to support the assumptions of health benefits of carotenoids are inconsistent, however, and mechanisms for such benefits have yet to be clearly articulated[6–8], leading to ongoing debate about the relevance of carotenoid pigments to immune or antioxidant defenses[6,7,9–12]. The showy carotenoid coloration of birds has been a particular focus of carotenoid research because diversion of carotenoids to ornamentation from immune or antioxidant function is hypothesized to form the basis of honest social signaling[3,13]. In this study, we used a white strain of domestic canary (*Serinus canaria*) to test the hypothesis that carotenoids play vital roles in avian physiology.

White recessive (WR) canaries carry a single-nucleotide polymorphism mutation that impairs the function of a carotenoid-transport protein (SCARB1), resulting in markedly low levels of carotenoids within tissue[14]. As a result, WR canaries grow white feathers, have white fat, and circulate colorless plasma (Fig. 1). They suffer severe retinol deficiency without supplementation[15], further demonstrating that only trace levels of carotenoids—some of which are major retinoid precursors—are physiologically available in the bodies of these canaries. Past research on SCARB1 mutants has been limited to laboratory mice[16], and to our knowledge, the physiological consequences of SCARB1-mediated extreme carotenoid reduction have not yet been explored with respect to immune or antioxidant function. Previous tests of the physiological benefits of carotenoids have manipulated dietary access to carotenoids, but such studies have yielded such complex and often equivocal outcomes even within well-studied bird species that the relevance of carotenoids to internal processes remains an open question[6,7,11]. Yellow (Y) canaries, which feature species-typical, carotenoid-rich tissues[14], and WR canaries are two color variants of the same breed of canary that differ phenotypically only in their tissue pigment content. Thus, comparison of the immune and antioxidant responses of carotenoid-rich Y canaries and carotenoid-deficient WR canaries presents a unique opportunity to assess the role of carotenoids in specific physiological processes. Well-controlled tests of the hypothesis that carotenoids are physiologically active and beneficial molecules in birds is critical to understanding both the evolution of carotenoid-based colored signals and the importance of these common dietary constituents to avian health.

We assessed the performance of Y and WR canaries on a variety of immune and antioxidant metrics in combination with physiological challenges, and we found no difference between the carotenoid-rich and carotenoid-deficient birds on any measurement. These results question whether carotenoid pigments directly boost physiological performance in the avian body.

## Results

**Experimental system**. We compared the physiological performances of WR canaries and Y canaries both under resting conditions and in response to immune and oxidative challenges. All birds received a dietary multivitamin supplement containing retinol (AviVita Plus, Avitec Bird Supplies), which prevented any confounding effects of retinoid deficiency between treatment groups. We performed all tests of physiological performance on canaries held in a long-term research colony at Auburn University.

**Levels of circulating carotenoids**. While the carotenoid content of liver, skin, feathers, and adipose tissue has previously been reported for both WR and Y canaries[14], we assessed levels of carotenoids in plasma samples from four WR and four Y birds. High-performance liquid chromatography revealed an average concentration of $0.74 \pm 0.36\,\mu g\,mL^{-1}$ ($\pm SD$) total carotenoids in circulation in WR canaries, compared to $20.31 \pm 21.26\,\mu g\,mL^{-1}$ in circulation in for Y canaries.

**Response to innate immune challenge**. To assess physiological performance in response to an innate immune system challenge, we first dosed WR and Y canaries with an intra-abdominal injection of bacterial lipopolysaccharide (LPS) from *E. coli* (O55:B5; $1\,mg\,mL^{-1}$ dissolved in phosphate-buffered saline (PBS); List Biological Laboratories Inc.). LPS is commonly used in songbirds and other species to trigger an acute phase innate immune response without the confounding effects of pathogen response to the host[17]. In our canaries, LPS injection induced a slight body temperature increase (average body temperature increase $\pm$ SEM over 8 h: $0.44 \pm 0.18\,°C$, $t = 5.01$, df = 25, $P < 0.001$) and decrease in mass (average mass lost over 8 h: $0.53 \pm 0.13\,g$, $t = 8.20$, df = 50, $P < 0.001$), but did not affect food consumption (average decrease in food consumption over 24 h: $0.014 \pm 0.37\,mg\,h^{-1}\,g^{-1}$, $t = 0.074$, df = 0.46, $P = 0.94$; Supplementary Fig. 1). 8 h after injection, we extracted a blood sample from birds to test total antioxidant capacity (TAC), oxidative burst response (a measure of innate immune cell production of pro-oxidants during defense against pathogens[18,19]; both peak and average response measured), and heterophil-to-lymphocyte (H:L) ratio (a broad measure related to immune stress in birds[20,21]). LPS injection tends to induce an oxidative stress challenge in birds[22], so our measure of TAC is related to antioxidant defenses present during immune activation. We observed

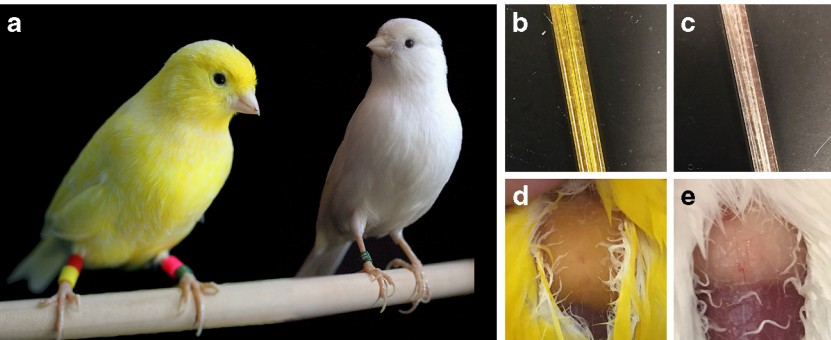

**Fig. 1** Comparisons of tissue coloration of yellow (Y) and white recessive (WR) breeds of canaries. **a** Y canaries grow bright yellow feathers; WR canaries grow pure white feathers. **b** Y canaries circulate yellow plasma and **c** WR canaries circulate colorless plasma. **d** The subcutaneous fat pads of Y canaries are yellow; **e** the fat pads of WR canaries lack yellow coloration

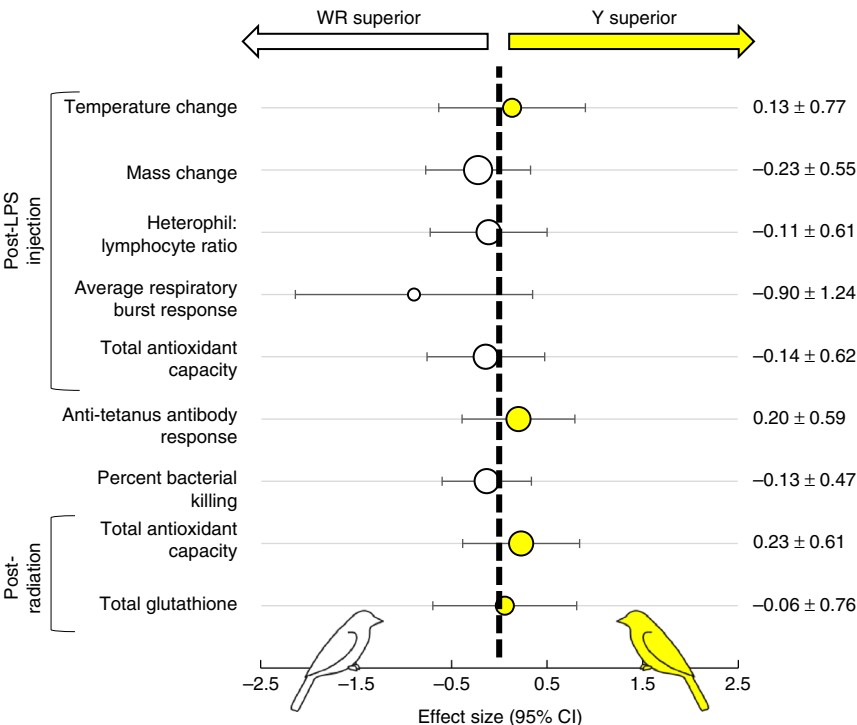

**Fig. 2** Plots of effect sizes comparing physiological responses of white recessive and yellow canaries. Each point and error bar represents an effect size (Hedges' *g*) and 95% confidence interval calculated from the average and standard error for a specific physiological response by WR or Y. The size of the point is proportional to the sample size; sample sizes are listed in Supplementary Table 1. Points shifted to the left of the dashed center line (white points) indicate a stronger response by WR canaries, while points shifted to the right of the center line (yellow points) indicate a stronger response by Y canaries. All confidence intervals include an effect size of zero, reflecting that there are no statistically significant differences in any response between WR and Y canaries. Values listed in the right column indicate effect size ± 95% confidence interval. The respiratory burst assays were performed on only a subset of birds (*N* = 11; see Methods), and the results should be interpreted with caution of that test should be viewed with caution

a slight increase in body temperature and decrease in body mass in response to LPS, independent of color type (see Methods). Importantly, we found no significant differences between carotenoid-deficient WR and carotenoid-rich Y canaries in any of the physiological parameters examined (analysis of variance (ANOVA); all *P* > 0.14; Fig. 2; Supplementary Fig. 2; Supplementary Table 1).

**Bacterial killing assay.** We further compared the innate immune responsiveness of WR and Y canaries using a bacterial killing assay (BKA) wherein we challenged fresh plasma samples with live *E. coli* (ATCC 8739; Microbiologics; Supplementary Fig. 1). We detected substantial inter-individual variation in bacterial killing capacity such that individuals of either color tended to kill either >90% or <10% of challenged bacteria relative to controls. We therefore assessed results using both ANOVA on percent bacteria killed and a binomial regression on likelihood to fully kill bacterial challenge, and we found no significant difference between WR and Y birds (ANOVA and binomial regression; *P* > 0.89; Fig. 2; Supplementary Fig. 2; Supplementary Table 1).

**Humoral antibody response.** To assess the effects of tissue carotenoid pigments on a different aspect of immune function, we activated the humoral immune system of WR and Y canaries with an intramuscular injection of human tetanus vaccine (TENIVAC, Sanofi Pasteur; Supplementary Fig. 1); this vaccine has been used successfully to induce antibody production in songbirds without causing lasting sickness[23]. Ten days after injection, we took a blood sample for use in quantifying each bird's humoral antibody response to tetanus. We used an established enzyme-linked

immunosorbent assay (ELISA) protocol[24,25] to quantify relative antibody responses of the birds, controlling for baseline response levels. All canaries had previously been vaccinated with the same vaccine at least 6 months before experimental injection and blood sampling, so the response we measured corresponded to a secondary antibody response. We found no significant difference in secondary anti-tetanus antibody responses between carotenoid-rich Y and carotenoid-deficient WR birds (ANOVA; *P* = 0.65; Fig. 2; Supplementary Fig. 2; Supplementary Table 1).

**Response to oxidative challenge.** Lastly, to assess the effects of carotenoid pigments on the capacity of canaries to cope with an oxidative challenge, we subjected the birds to low-dose (50 rads) X-irradiation, which induces pro-oxidant production in vivo. Low-dose ionizing radiation is a new method of experimental oxidative challenge that has the advantage of inducing system-wide increases in pro-oxidants without causing clinical disease symptoms or organ-specific dysfunction (as may be observed in chemical challenges[26]). Twenty-four hours after irradiation (Supplementary Fig. 1), we extracted a blood sample for anti-oxidant analyses. We measured TAC in isolated plasma and we quantified the levels of one endogenous antioxidant (total glutathione; Cell BioLabs) in red blood cells. We used total glutathione levels as an indicator of whether WR birds may be compensating for their lack of carotenoids by upregulating the production of endogenous antioxidants. Exposure to X-irradiation induced physiological changes as indicated by higher average TAC values for individuals after radiation compared to after LPS injection (paired *t* test; *t* = 4.18, df = 29, *P* < 0.0001). However, we observed no significant differences

between Y and WR birds in TAC or total glutathione (ANOVA; $P = 0.46$ and $P = 0.91$, respectively; Fig. 2; Supplementary Fig. 2; Supplementary Table 1).

## Discussion

As measured by a variety of immune and antioxidant parameters and in response to different physiological challenges, WR canaries with severely depleted tissue carotenoid levels performed identically to Y canaries with species-typical, high levels of carotenoids. While we cannot rule out the possibility that the very low levels of tissue carotenoids in WR canaries[14] still enabled key physiological function, the capacity of canaries with very low levels of circulating carotenoids to cope with immune and oxidative challenges that carotenoids directly boost immune and antioxidant performance in birds. Moreover, few avian taxa (even those without carotenoid-based coloration) have been found to have plasma carotenoid levels as low as those of WR canaries (see Methods)[27], which suggests that essentially all birds have access to sufficient dietary carotenoids to meet physiological needs.

By providing retinol as a supplement to both WR and Y birds, we prevented retinol insufficiencies in the carotenoid-free birds and thereby isolated tissue carotenoid content as the only functional difference between the two strains. Equalizing retinol access between the two groups allowed us to test the potential direct benefits of intact carotenoids while controlling for the indirect effects of carotenoids as retinoid precursors, which is often not possible in natural systems. While the physiological benefits of retinoids are widely accepted, the relationships between retinol, internal carotenoids, coloration, and physiological quality are not often tested and the positive effects of retinol are rarely considered in studies of carotenoid signaling. A recent meta-analysis indicated that carotenoid and retinol levels tend to be correlated in avian plasma[28], which suggests that some positive correlations between internal carotenoids and physiological performance may arise from the beneficial effects of retinol, not carotenoids per se.

Currently, there is no single, widely accepted alternative hypothesis to explain associations between carotenoid pigments, carotenoid-based colored signals, and individual condition. One important follow-up to our study is to more closely examine the relationship between retinol, its physiological benefits, and carotenoids. Indeed, the vitamin A (retinol)-redox hypothesis offers a biochemical explanations for how carotenoid signals may vary along with the same internal processes related to retinol processing and beneficial function[29], though the predictions of this hypothesis remain to be demonstrated experimentally[28]. Similarly, other recent hypotheses consider that the honesty of carotenoid-based coloration may be maintained through index signaling mechanisms that do not require a physiological cost of coloration or benefit of carotenoids[30,31]. For example, if the full expression of carotenoid signals depends on the proper function of a core cellular process then coloration would be inexorably linked to the myriad of internal processes also dependent that process[32–34]. Indeed, the process of converting dietary carotenoids into ornamental carotenoids, which occurs in many species with carotenoid-based coloration, may itself be a key factor in signal expression[29,35,36].

We argue that the key to differentiating among such hypotheses lies in novel systems like the WR canaries that allow us to separate the effects of carotenoids from other physiological processes. In studies of natural systems, it is often difficult to distinguish whether relationships detected between carotenoid pigments and internal physiological function are causal or correlational. New systems that take advantage of our growing understanding of the genetics underlying carotenoid-based coloration[14,37,38] will be critical for providing new perspective on long-standing questions in carotenoid signaling and function.

It will be important to expand our observations of canaries to other avian and vertebrate systems to further test hypotheses of direct carotenoid benefits to physiological processes. Given that the panel of measurements we performed on the birds was inherently limited by the scope of our study, further investigation of WR canaries themselves will be important to more fully characterize their physiology, particularly in relation to redox processes (e.g., measuring lipophilic estimate of antioxidant capacity[39]). Future studies must also rule out the possibility that WR birds may compensate for their lack of carotenoids through other means, such as by increasing production of endogenous antioxidants (though we detected no differences in the levels of one antioxidant, glutathione). In addition, we advocate for the development and testing of additional vertebrates with genetic knockdown or knockout SCARB1 to serve as models in carotenoid research to expand our findings beyond a single songbird taxon. We further urge future studies on both captive and natural systems to consider the interactions between retinol, retinol precursor carotenoids, and physiology when drawing conclusions about the role of carotenoid pigments in condition-dependent ornamentation. That the SCARB1 knockdown WR canaries can thrive in spite of their severe carotenoid deprivation challenges the hypothesis that carotenoids provide direct physiological benefits to the avian body.

## Methods

**Animal husbandry.** We performed our tests on a long-term research colony of after-hatch-year color-bred canaries held at the Auburn University Avian Research Laboratory 1 in Auburn, AL. All experimental and husbandry procedures were approved by the Auburn University Animal Care and Use Committee (PRNs 2014-2465, 2014-2499, 2015-2724, and 2015-2789). Both WR and Y canaries had been maintained in our aviary for at least 1 year after acquisition from large, long-term colonies maintained by aviculturists in the United States; both WR and Y canaries had been born and raised under identical conditions. The WR and Y canaries assessed in this experiment are both of the same general breed ("color-bred canaries"), and differ only in their carotenoid phenotype, which has been traced to a mutation in the *SCARB1* gene[14]. WR and Y canaries interbreed successfully and are crossed to maintain healthy genetic diversity in aviculturist breeding flocks. The WR gene inherits in a Mendelian recessive manner, and while we were unable to perform the two generations of crosses needed to produce full-sibling WR and Y birds for this experiment, we did perform F1 crosses between WR and Y birds not included in this study; offspring of such crosses are phenotypically indistinguishable from both their parents, except for their full expression of yellow plumage. The average Fst value between our WR and Y birds is likely much lower than that reported previously because a variety of wildly different canary breeds (e.g., canaries bred for aberrant body posture) contributed to the genetics of the yellow canaries used in that experiment[14], while our WR and Y birds are of the same breed and have been raised in the same aviculturist breeding colonies for generations. Functionally, we predict our WR and Y birds to differ only in their alleles for the carotenoid-absorption *SCARB1* gene.

We held the canaries on a carotenoid-controlled diet of mixed canary seed (predominantly canary grass seed, mixed with rapeseed and thistle; All Natural Canary Blend, Jones Seed Company) coated with a carotenoid-free vitamin powder (AviVita Plus, Avitec Bird Supplies). The vitamin powder includes retinol, provided within the range recommended for both Y and WR birds to circulate healthy retinol levels[15], which is critical because WR birds cannot absorb retinol precursor carotenoids. This diet contains a moderately low quantity of carotenoids ($<20 \mu g\ g^{-1}$, primarily lutein and β-carotene with low levels of zeaxanthin[40,41], which is sufficient for the birds to breed, molt, and fully color their feathers (in the case of Y canaries) successfully. The values of circulating carotenoids we report in Y canaries, along with their tissue carotenoid concentrations[14], are comparable to those found in wild songbirds with carotenoid coloration[27,42], and we therefore do not expect our Y birds to have been unusually limited in carotenoid access. Our primary goal was to avoid supplementing the birds with large amounts of carotenoids that might lack biological relevance to the varied diet of wild songbirds[43]. Birds were sexed by vent morphology and behavior (singing, egg-laying) in previous breeding seasons.

**Statistics.** We performed ANOVAs to test for significant differences in response measurements based on color type (Y vs. WR), sex (male vs. female), or the interaction of sex and color type; we controlled for the effects of sex because measurements of internal performance have often been found to differ between the

two sexes[44–46]. However, given that sex and the interaction of sex and color played little role in our results (Supplementary Table 1), we focus on the comparison between WR and Y birds.

For LPS results analyses, we also performed two-tailed paired $t$ tests to assess whether LPS injection significantly affected mass, food consumption, or temperature across both types of birds (see below). We similarly used a two-tailed paired $t$ test to assess whether the difference in mean TAC values between post-radiation and post-LPS injection measurements was different than zero. For BKA result analyses, we tested for significant differences in WR and Y performance using binomial regression in addition to ANOVA (see below). All statistical analyses were performed in R (version 3.2.3)[47].

**Plasma carotenoid analysis.** Prior to the commencement of the main experiments, we performed carotenoid content analyses on plasma samples taken for previous experiments from four WR and four Y canaries in our colony and frozen at −80 °C for <6 months. Frozen samples were shipped to Washington University School of Medicine (St. Louis, MO, USA), where they were analyzed in the lab of Dr. Joseph Corbo. To extracted carotenoids from plasma samples, we added 250 μL of ethanol followed by 250 μL of hexane:tert-methyl butyl ether (1:1 vol:vol) to 15–20 μL plasma samples, then we centrifuged the samples at 10,000 rpm for 3 min in bench top centrifuge, collected all of the supernatant, and dried it under a stream of nitrogen. We resuspended the plasma extracts in 120 μL of methanol:acetonitrile 1:1 (vol:vol) and injected 100 μL of this suspension into an Agilent 1100 series HPLC equipped with a YMC carotenoid 5.0 μm column (4.6 mm × 250 mm, YMC). We separated the carotenoids with a gradient mobile phase consisting of acetonitrile:methanol:dichloromethane (44:44:12) (vol:vol:vol) through 11 min, then a ramp up to acetonitrile:methanol:dichloromethane (35:35:30) from 11 to 21 min followed by isocratic conditions through 35 min. We heated the column to 30 °C, and the maintained a flow rate of 1.2 mL min$^{-1}$ throughout the run. We monitored the samples with a photodiode array detector at 400, 445, and 480 nm, and carotenoids were identified by comparison to authentic standards or published accounts[14].

**LPS challenge.** In July–August 2016, we ran a bacterial LPS (see below) challenge on WR and Y canaries (Supplementary Fig.1). Briefly, working with 2–6 individuals per day, we moved birds from their long-term colony cages to individual 12 × 16 × 16 in. cages. Twenty-four hours prior to injection, we provided each bird with a known quantity of canary seed mix; we then collected all remaining seeds and husks from each bird's cage immediately prior to LPS injection (after about 24 h of consumption) to calculate resting food consumption for each bird (adjusting for each individual's baseline mass and the exact hours elapsed between measurements, resulting in units of mg seed h$^{-1}$ g per body mass$^{-1}$). We repeated this process for birds for the 24 h immediately after LPS injection to test for any changes in food consumption following the challenge. We also measured the mass and temperature (using a Leaton Digital Thermocouple Thermometer inserted ~1 cm into the vent) of birds immediately before and 8 h after LPS injection. LPS injection has been found to decrease mass, body temperature, and food consumption in many species[17].

On the morning of the LPS challenge, we injected each experimental bird intra-abdominally with 1 mg mL$^{-1}$ lipopolysaccharide from *E. coli* (O55:B5; List Biological Laboratories Inc.) dissolved in PBS. Eight hours post-injection, we collected blood samples from each experimental bird by puncturing the wing vein with a 26-gauge needle. We dispensed approximately 30 μL of blood into a heparin-coated 1 mL capped tube for use in respiratory burst assays; we spread an additional small drop of blood evenly on a microscope slide for cell count analyses; and, we collected a final 150 μL of blood in two heparinized capillary tubes, which we immediately centrifuged to extract plasma and red blood cell samples for storage at −80 °C until further analysis.

**Respiratory burst assay.** We assessed the respiratory burst potential of immune cells contained in the fresh whole blood sample taken 8 h post-LPS injection. Respiratory burst is the process by which immune cells produce large amounts of reactive oxygen species (ROS) to disrupt potential pathogens[19]; however, because ROS cannot be "aimed" at specific targets, there is the potential for respiratory burst to also damage host tissue. As a consequence, one of the main roles for carotenoid pigments as antioxidants and immune boosters has been proposed to be defense against damage to self during respiratory burst, by boosting immune cell development and/or by quenching ROS as they are produced. Interestingly, these possible roles of carotenoids in the process of respiratory burst lead to contradictory predictions: if carotenoids boost immune cell development and function, then we would expect Y birds to have increased respiratory burst response relative to WR birds; however, if carotenoids provide benefits by rapidly quenching ROS, then Y birds should have lower levels of ROS during the respiratory burst response. Some in vitro studies on mammalian cell cultures indicate increased respiratory burst in the presence of carotenoids[5]. In previous assessments of respiratory burst in songbird whole blood samples, however, no effects of dietary carotenoid supplementation have been found[18,48]. Here, we use comparable methods to test whether a near-complete absence of carotenoids affected the ability of immune cells contained within whole blood to mount a strong respiratory burst response.

We followed the kit protocol of the Analysis By Emitted Light Cell Activation Kit with Pholasin (Knight Scientific), modified for use without reagent injectors; this protocol uses a chemiluminscent reagent that releases light upon reaction with ROS or other free radicals. Briefly, we dispense LPS into whole blood samples mixed with kit reagents and measure the luminescence that results as white blood cells respond to the stimulus with ROS production. The result is a steep increase in luminescence immediately after LPS dispensation, then a gradual drop-off as respiratory burst response slows. We performed the assay in duplicate for each individual within 2 h of the post-LPS blood draw. Due to unexpected loss of efficacy of reagents, we measured respiratory burst in only a small subset of birds (6 WR and 5 Y). For each sample, we subtracted the baseline values from the experimental values to calculate the net increase in luminescence due to LPS-stimulated respiratory burst. We examined peak response (maximum luminescence; corresponds to maximum ROS production immediately after LPS exposure) as well as average luminescence over the 60-s post-injection interval (corresponding to duration of sustained response).

**Heterophil-to-lymphocyte ratio.** The ratio of heterophils (the avian analog to mammalian neutrophils) to lymphocytes, while not a definitive reflection of any one immune parameter, has previously been cited as a general indicator of overall immune activation in birds[21,49], and has specifically been related to health[50] and carotenoid-based coloration[20] in songbirds. To assess H:L ratio in WR and Y canaries, we first fixed and stained (Fisher HealthCare PROTOCOL Hema 3 Fixative and Solutions) slides of whole blood smears (collected 8 h post-LPS injection; see above) to visualize cell types. Using standard techniques for avian blood cell counting[20] and type identification, we counted at least 10,000 total cells per individual (estimated based on total slides viewed and average cell counts per three representative slide views); we examined multiple regions of each blood smear to gain an accurate reading of blood cell parameters across the slide, and we only measured views of cells evenly spread in a single layer. We counted total cells, number of heterophils, number of lymphocytes, and number of thrombocytes; no other cell types were present across all individuals. We divided total number of heterophils by total number of lymphocytes to calculate the H:L ratio for each individual.

**Total antioxidant capacity.** We measured TAC in plasma samples using the TAC kit (OxiSelect$^{TM}$ TAC Assay Kit, Cell BioLabs) according to the provided protocol. This assay relies on the tendency of antioxidants to reduce copper (II) to copper (I), creating a color change in the solution that can be compared to a known uric acid standard dilution. After preliminary testing, we established that a 1:4 dilution of canary plasma yielded best results within the provided standard curve. We diluted 5 μL of plasma in 15 μL of PBS in duplicate for each individual. Results are reported in units of μM copper reduction equivalents.

**Vaccination.** In August 2016, we extracted 75 μL of blood (pre-vaccination sample) from each experimental canary and then immediately injected each bird intramuscularly with 100 μL of pharmaceutical-grade tetanus vaccine (2 lf of tetanus toxoid; also contained 2.7 lf of diphtheria toxoid; TENIVAC, Sanofi Pasteur, France), dispensing 50 μL into each breast muscle. Ten days later, we extracted a second 75 μL blood sample (the post-vaccination sample). All blood samples were kept on ice and centrifuged to extract plasma from red blood cells for storage at −80 °C until analysis. A subset of plasma from the pre-vaccination sample was used immediately in BKAs (20 μL; see below), while the remainder of plasma was frozen at −80 °C until further analysis.

In September 2016, frozen pre- and post-vaccination plasma samples (~15 μL each) were transported to Lund University (Lund, Sweden) for antibody analysis in the lab of Dr. Dennis Hasselquist. Antibody responses were assessed using ELISA methods developed in several songbird species for use quantifying anti-tetanus antibodies in avian plasma[24,25]. Canary plasma samples were diluted 1:1,000, as preliminary tests indicated that this dilution produced measurable responses falling within the bounds of measurements from a positive control used on all plates (serial dilutions of plasma from great tits (*Parus major*) with known strong responses to tetanus). For the ELISA, we first incubated 96-well ELISA plates overnight with tetanus toxoid (Statens Serum Institute, Copenhagen, Denmark), then blocked the plates with a dilution of 3% powdered milk in 0.01 M PBS. We diluted each individual's plasma sample in a diluent of 1% powdered milk in 0.01 M PBS, and we incubated the plates with samples in duplicate. Each plate also contained a duplicate serial dilution of the positive control sample to use as a reference to standardize among plates. After sample incubation and wash (with PBS and Tween-20 in a BioTek ELx50 ELISA washer), we added a secondary rabbit anti-songbird immunoglobulin antiserum developed previously.[25] After another incubation and wash, we added a commercial peroxidase-labeled goat anti-rabbit antiserum (cat. no. A6154, Sigma; Sigma-Aldrich, Sweden), and incubated and washed the plate one more time. Finally, we added the peroxidase substrate (2 2,2-azino-bis-3-ethylbenzthiazoline-6-sulfonic acid; cat. no. A1888, Sigma) and peroxide, and we immediately transferred each plate to a kinetics plate reader (BioTek EL 808). Plates were read at 30-s intervals for 16 min using a 405 nm wavelength filter. From the results, we calculated the slope of the substrate conversion over time, measured in units of milli-optical density min$^{-1}$. Each measurement was adjusted for among-plate variation (according to between-plate

variation in the measurements of the great tit control samples), duplicate measurements of each individual were averaged (excluding a small number of individuals where duplicates had >10% intra-individual variation), and each individual's average was log-transformed for linearity. The net "antibody response" for each individual was calculated as the difference between pre- and post-vaccination measurements.

**Bacterial killing assay.** On a subset of plasma extracted pre-vaccination in August 2016, we performed a modified microplate-based plasma BKA[51]. Briefly, the day prior to assays, we reconstituted a bacterial pellet (*E. coli*, ATCC 8739; Micro-biologics Epower) in 40 mL of warm, sterile PBS, and assessed colony-forming units (CFUs) using standard agar plating protocols under sterile conditions in a laminar flow hood. Based on calculated CFUs, we diluted the stock bacterial solution to obtain a working solution of ~$1 \times 10^5$ CFUs for assays. We extracted plasma from each capillary tube under sterile conditions and retained 20 μL of each individual's plasma in a single, sterile 1.5 mL pop-cap tube for BKA use (the remaining plasma was frozen at −80 °C). We diluted plasma with 80 μL of sterile PBS, vortexed the mixture thoroughly, then plated 20 μL of each diluted sample to a "negative plasma control" well on a 96-well, round-bottomed microplate. These negative plasma controls served as tests for blood sample contamination. We repeated these methods for one sterile fetal bovine serum (FBS) control, which served as a point of comparison for bacterial growth in the presence of plasma without bacterial killing components. This FBS control was used as the positive control for comparison with experimental samples because FBS readings represent bacterial growth in the presence of plasma with little to no bacterial killing components.

We added 8 μL of our prepared bacterial working solution to each individual's remaining 80 μL of diluted plasma, vortexed the mixture, then incubated each tube at 37.4 °C for 30 min. Eighty microliters of diluted FBS and 80 μL of sterile PBS were each also mixed with 8 μL of bacterial working solution for positive controls. After incubation, we vortexed samples again and plated 20 μL of each sample in triplicate on the microplate. We added 125 μL of sterile tryptic soy broth to each well, mixed gently using a multichannel pipette, and read the plate at a wavelength of 600 nm (for baseline absorbance). We then covered the plate and incubated on a rocker for 12 h at 37.4 °C before performing a second reading at 600 nm.

These methods were determined after several pilot analyses involving different concentrations of plasma, bacteria, or whole blood samples, and different incubation lengths. We selected the above dilutions because we found them to best isolate a range of results among individuals; other dilutions tended to result in either no bacterial growth or no evidence of killing in experimental wells. To assess the results, we first calculated an average net absorbance for each individual and control by subtracting baseline (time 0) values from 12-h values to remove any underlying differences in absorbance between samples. We then eliminated any outlying values in each individual's triplicate of absorbance readings that differed more than 10% from the other two values. Finally, we averaged the remaining net absorbance values for each individual and control. We divided this final net absorbance for each individual by the final net absorbance of the positive control on the same microplate to obtain a value for percent difference in absorbance—or, more specifically, percent difference in bacterial growth—between samples and control. This percent-bacteria-killed value for each individual was used in further analyses.

However, the results were unusual in that individuals tended to consistently either completely kill (<10% bacterial growth compared to FBS-positive controls) or completely fail to kill (>90% bacterial growth compared to FBS-positive controls) their bacterial challenge, indicating large inter-individual variation in performance. As such, we performed a binomial regression analysis in addition to ANOVA to assess statistical patterns in the data. For the binomial analysis, we excluded five data points with percent-bacteria-killed values between 10 and 90% (i.e., those with partial killing) so that all remaining individuals could be categorized as having fully killed or fully failed to kill their bacterial challenge.

**Radiation challenge.** In December 2016, we dosed experimental canaries with 50 rads of X-irradiation using a PRIMUS (Siemens) linear acceleration at the Radiology Department of Clinical Science in the Auburn University College of Veterinary Medicine. Each bird was secured in a brown paper bag during exposure. The dose was based on a low dose previously reported to induce an oxidative damage increase in two other songbird species[52]. We observed no change in mass or clinical signs of distress after the procedure. Twenty-four hours post-irradiation, we extracted a 150 μL blood sample and centrifuged to isolate plasma from red blood cells; we froze both at −80 °C for further analysis.

On plasma samples extracted post-irradiation, we performed TAC measures using identical methods as described above. On red blood cell samples extracted post-irradiation, we assessed the total glutathione concentration using the Total Glutathione Assay Kit (Cell BioLabs). Briefly, we mixed 10 μL of red blood cell pellet from each individual with 40 μL of a 5% metaphosphoric acid solution; after centrifuging the mixture, we extracted the supernatant and diluted it 1:250 (optimized according to pilot assay results) in an assay buffer containing 0.5% metaphosphoric acid. We then followed the kit protocol as listed; the procedure converts any reduced glutathione to oxidized glutathione, which is then measured through a chromogenic reaction in order to assess total glutathione levels without

regard to specific oxidation state. Our goal was to gain an indicator of an endogenous antioxidant level in both WR and Y canaries in order to gauge whether carotenoid-free birds may have upregulated other antioxidants to compensate for the absence of carotenoids. We assessed each individual's total glutathione levels in duplicate and compared them to a standard curve of known total glutathione concentration.

**Data availability.** The data that support the findings of this study are available from the corresponding author upon reasonable request.

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

## Acknowledgements

We thank J. Corbo for assistance with carotenoid analyses, A. Hegemann and C. Birberg for assistance with vaccine antibody assessment, G. Almond for irradiation procedures, and R. Amin for providing the laboratory equipment used in oxidative burst assessment. Members of the Hill and Hood labs and Auburn University undergraduates assisted with live animal procedures. Oxidative stress procedures were partially funded by a National Science Foundation grant (1453784) to W.R.H. D.H. was supported by grants from the Swedish Research Council (VR; 621-2013-4357, 2016-04391).

## Author contributions

R.E.K. and G.E.H. conceived the project. R.E.K. performed animal husbandry, live animal procedures, and most laboratory analyses, with assistance from A.N.K. on oxidative stress procedures and from Y.Z. and W.R.H. on radiation protocols and analyses. D.H. performed tetanus antibody analyses. M.B.T. performed carotenoid analyses. R.E.K. and G.E.H. wrote the manuscript and all authors contributed to revisions.

## Additional information

**Competing interests:** The authors declare no competing financial interests.

