## [Peer Review File · Nature Communications]

Reviewers' comments:

Reviewer #1 (Remarks to the Author):

The described study has been carried out in two different strains of canary, a wild type and a knockout SCARB1. SCARB1 individuals are not able to absorb enough carotenoids to pigment their feathers, skin or plasma. Since they also lack in retinol precursors, which are also carotenoids, they must be supplemented with retinol to survive. Birds of both lines have been exposed to a series of immune and oxidative challenges and since they did not differ in any of the used markers, the main claim of the study is that carotenoids are not able to boost immune or antioxidant performance. The experimental approach is novel because no other studies using knockout SCARB1 models tested the effect on immune system responsiveness and antioxidants protections. Findings are of great interest as they challenge the general perspective of carotenoids, considered to be able to promote the activity of the immune system. About the antioxidant function, their role has already been scaled down (e.g. Costantini and Moller 2008), but we still lack clear conclusions, thus studies like the present one are needed. The paper is very well and clearly written, the images and tables provided are of high quality. I have some comments that I am sure the authors can easily address.

The parameters used to assess the antioxidant actions of carotenoids lack of a measure of oxidative damage. The molecular network controlling the homeostasis of the redox state is very complex and flexible. The mitochondria can control, to a certain extent, both the amount of ROS that can be produced and the antioxidant enzymes such as GPX, CAT and SOD that buffer ROS inside the cell. It would have been important to characterize at least one of these enzymes and provide a measure of oxidative damage. Without this information I think that the claim that carotenoids have not antioxidants effect at all should be toned down. I have some comment also about how TAC has been measured. I could not find what the kit used was able to detect (this info should be specified in the methods). Looked at the kit explanation available on line for commercial purpose it seems to measure only acid uric, a waste product of protein. How carotenoids and uric acid should be related? What is the expected effect of carotenoids on the non-enzymatic antioxidant barrier in this experiment?

Both strains of canary have been provided with retinol during the entire experiment and retinol has both antioxidant and pro-immunity properties. The authors stated that (lines 101-103) this supplementation "allowed them to unambiguously separate the potential direct benefits of intact carotenoids from the indirect effects of carotenoids as retinol precursors". This explanation should be further extended because it is not very clear. In which specific cases these differential benefits have been unambiguously separated? Could it be possible that the administered retinol was enough to protect the organism from oxidative insult and to boost the immune system? I could not find the concentration of retinol provided with food.

The knockdown line retains traces of carotenoids in the body (line 25). Checking the concentration of carotenoids in the cited reference (Ref 10), the liver of white canaries contain about 2 $\mu\text{g}\cdot\text{g}^{-1}$ of carotenoids instead of the typical 13 $\mu\text{g}\cdot\text{g}^{-1}$ of yellow canaries

(Toomey et al. 2017; please note that other coloured species can have much less carotenoids in the liver, e.g. house finches, McGraw et al. 2006). Since the liver is a fundamental organ for carotenoid metabolism it would be important to explain why these “traces” of carotenoids are negligible and their physiological action can be excluded. Would it be possible that low traces are enough for fulfilling antioxidants and/or immune enhancing functions and that surplus is used by yellow type to colour skin and feathers? I couldn't not find any data about carotenoid concentration in the blood, is this known?

The dose of carotenoids provided with the food was low, as stated by the authors in line 240. Would it be possible that providing them a greater amount of dietary carotenoids would have led to different results? I suggest addressing this issue in the discussion.

More information about the respiratory burst response (line 294-305) should be provided because this is not a very common technique used to assess the performance of the immune system. In particular, expected results in relation to the action of carotenoid should be clearly explained. Is it that carotenoids would increase the production of reactive oxygen species because they boost the immune system, thus the respiratory burst of heterophils? Please explain. Moreover, when discussing the results of the respiratory burst response it should be specified that they cannot be considered conclusive because of the very low sample size (6WR and 5 Y of different sexes, Line 311 and Table 1 extended data). Since it almost missed the significance this could be due to low power, thus further studies should be suggested to test if the higher respiratory burst observed in white canaries is really not significant.

Birds have been exposed to several treatments and several bleeding. A timetable of the study design (now there are some general indications about when a particular treatment has been done, but it is difficult to follow) and the amount of blood taken each time should be reported as repeated bleeding could interfere with several physiological pathways and individual condition.

Reviewer #2 (Remarks to the Author):

The authors use a novel system – the white strain of the domestic canary – to test the long held view that carotenoids enhance immune function or antioxidant defenses. The authors use a diversity of approaches to evaluate immune and antioxidant defenses in white canaries (with trace circulating & tissue carotenoids) and yellow canaries (with normal amounts of carotenoids). Their tests are comprehensive and they find no hint of a difference between the two strains. The experiment is elegant and timely – the authors should be commended for providing a robust and serious challenge to the widely touted assumption that carotenoids play important roles in key physiological functions. This is the kind of study that will cause many of us to go back to old published work and have a very serious think about alternative interpretations of the reported results.

The results of this study are novel, they will have undoubtedly a very high impact on the

field, and they will be of wide interest to evolutionary ecologists and the wider field. The experiment is simple – which is its main strength. And the authors utilized fully four different approaches to test for differences between mutant and yellow strains – the study's other main strength. The statistics are fine and sample sizes are large enough to have detected any trends if there were any. There is no need for overly complex stats for this particular study.

I have no major concerns with the paper – the writing was clear and concise. One question that comes to mind, however, is why has it been the case that the hypothesis that carotenoids play a direct role in immune or antioxidant function has been such a pervasive feature of honest signalling theory? Is it just that we *wanted* it to be the case for facilitating a just-so story about signal costs? Or has any putatively positive evidence for the hypothesis simply been interpreted incorrectly? Perhaps this paper is not the platform for such discussions, but the question might be worth exploring in more detail.

James Dale

Reviewer #3 (Remarks to the Author):

The manuscript describes a study comparing the response of two breeds of domesticated canary to a variety of immune and oxidative challenges. The experimental challenges are start of the art, in particular the breadth they cover.

While the results are clear and conclusive, various aspects of the experimental design are relevant for its inference, which in my view is more restricted than what the authors imply. Doubt that carotenoids per se are important antioxidants and immune stimulants dates back a decade. In the evolutionary ecology literature, the thinking has become more sophisticated after the initial proposal that the dual role of carotenoids as antioxidants and colourants directly enforces honesty on sexual signals. Various authors have pointed out that the role of carotenoids as antioxidants is possibly not large in birds (as also shown by references 6 and 7 in the manuscript). Indeed, the last 10 years of research have painted a more complex picture. Nonetheless, I agree with the authors that the field remains fairly muddled, despite some recent meta-analyses and reviews. Thus, it is critical that seminal contributions to high profile journals clarify the field, and make no unsubstantiated claims: in order to test for nutraceutical benefits, various realistic levels of supplementation against a homogenous, but natural, diet background of relevant study animals is critical. To test the proposed evolutionary ecological role of carotenoids in signalling, ecological and evolutionary relevance is important, that is natural conditions and carefully controlled life-history. In my view, this manuscript does not decisively contribute to either of these fields. Yellow canaries are a domesticated breed, selected for high carotenoid incorporation into the plumage. Since plasma carotenoid levels and carotenoid processing machinery can also evolve rapidly in response to selection for ornamentation (Simons et al. 2014. Carotenoid-Dependent Signals and the Evolution of Plasma Carotenoid Levels in Birds. *Am Nat* 184:741–751) it is critical to know what other genetic and physiological changes have occurred in the canary breeds, that were both a result of independent artificial selection

processes. For example, white canaries might have compensating mutations to deal with absence of carotenoids, yellow canaries might have been selected to be insensitive to physiological effects of dietary carotenoids.

While it is clear that both canary breeds respond similarly, although it seems the caveat is that this result depends on the diet containing sufficient retinol (vitamin A) – the specific deficiency that results in the white canaries as a result the mutation in SCARB1.

The white canaries critically depend on a dietary source of retinol (vitamin A). In natural situation, vitamin A levels are positively related to carotenoid plasma levels (Simons et al. 2014. An appraisal of how the vitamin A-redox hypothesis can maintain honesty of carotenoid-dependent signals. *Ecol Evol* 5:224–228), and this should be discussed as a potential experimental artifact or explanation for correlative patterns that are found in natural situations.

Additionally, this is not a true experimental study, in order to conclude whether carotenoids boost immune or antioxidant defenses, both breeds should have been exposed to diets low and high in carotenoids and low and high in retinol. Non-experimental captive studies in general are not suitable to demonstrate allocation trade-offs and the evolutionary ecological relevance of the study is difficult to assess.

The dietary composition is not very well justified, what can be considered a natural dietary carotenoid content for domestic yellow or white canaries? This is an important consideration, and I would have expected more of a discussion of this issue given the previous – and important - contributions by the authors on the importance of doing so (Koch et al. 2016. The Importance of Carotenoid Dose in Supplementation Studies with Songbirds. *PBZ* 89:61–71)

No information on breeding and rearing conditions of canaries, this is important given the long-lasting effects of such conditions (Evans SR, Sheldon BC. 2012. Quantitative Genetics of a Carotenoid-Based Color: Heritability and Persistent Natal Environmental Effects in the Great Tit. *Am Nat* 179:79–94.)

Reviewer #1 (Remarks to the Author):

The described study has been carried out in two different strains of canary, a wild type and a knockout SCARB1. SCARB1 individuals are not able to absorb enough carotenoids to pigment their feathers, skin or plasma. Since they also lack in retinol precursors, which are also carotenoids, they must be supplemented with retinol to survive. Birds of both lines have been exposed to a series of immune and oxidative challenges and since they did not differ in any of the used markers, the main claim of the study is that carotenoids are not able to boost immune or antioxidant performance. The experimental approach is novel because no other studies using knockout SCARB1 models tested the effect on immune system responsiveness and antioxidants protections. Findings are of great interest as they challenge the general perspective of carotenoids, considered to be able to promote the activity of the immune system. About the antioxidant function, their role has already been scaled down (e.g. Costantini and Moller 2008), but we still lack clear conclusions, thus studies like the present one are needed. The paper is very well and clearly written, the images and tables provided are of high quality. I have some comments that I am sure the authors can easily address.

The parameters used to assess the antioxidant actions of carotenoids lack of a measure of oxidative damage. The molecular network controlling the homeostasis of the redox state is very complex and flexible. The mitochondria can control, to a certain extent, both the amount of ROS that can be produced and the antioxidant enzymes such as GPX, CAT and SOD that buffer ROS inside the cell. It would have been important to characterize at least one of these enzymes and provide a measure of oxidative damage. Without this information I think that the claim that carotenoids have not antioxidants effect at all should be toned down.

RESPONSE: We attempted several methods that measure various proxies of oxidative stress/damage in the plasma—dROMs, protein carbonyls, 4-hydroxynonenol, and reduced/oxidized glutathione—and we were unsuccessful in obtaining reliable data due to several reasons. We were limited in the plasma volume available to us due to the small quantity of blood that we were able to collect from our 25g birds, and the assays did not seem to be sensitive enough to reliably detect these measures at such quantities; although we tried to troubleshoot and optimize the assays, we ultimately ran out of plasma. As such, we are unable to report any reliable estimates of oxidative damage in our birds. Total antioxidant capacity was the next best measure we could use, within our remaining budget and plasma samples, to examine overall antioxidant performance. While we did measure total glutathione in an attempt to gauge whether there may be other differences in endogenous antioxidants due to WR “compensation,” we agree that measuring GPX and/or the other enzymes would be an important next step to research in this system.

I have some comment also about how TAC has been measured. I could not find what the kit used was able to detect (this info should be specified in the methods). Looked at the kit explanation available on line for commercial purpose it seems to measure only acid uric, a waste product of protein. How carotenoids and uric acid should be related? What is the expected effect of carotenoids on the non-enzymatic antioxidant barrier in this experiment?

RESPONSE: The TAC kit we used is the Oxiselect™ Total Antioxidant Capacity (TAC) Assay Kit from Cell BioLabs. We have added greater detail regarding the kit used and its methodology to the Methods section as suggested. This assay relies on the tendency of antioxidants to reduce copper (II) to copper (I), creating a color change in solution that can be compared to a known uric acid standard dilution.

Both strains of canary have been provided with retinol during the entire experiment and retinol has both antioxidant and pro-immunity properties. The authors stated that (lines 101-103) this supplementation “allowed them to unambiguously separate the potential direct benefits of intact carotenoids from the indirect effects of carotenoids as retinol precursors”. This explanation should be further extended because it is not very clear. In which specific cases these differential benefits have been unambiguously separated? Could it be possible that the administered retinol was enough to protect the organism from oxidative insult and to boost the immune system? I could not find the concentration of retinol provided with food.

RESPONSE: We have reworded the sentence in question, and we now further discuss the implications of our findings with respect to retinol in the discussion. We agree that retinol may very well be playing an important role in immune and/or antioxidant processes. While we were not able to use laboratory analyses to assess the exact quantity of retinol in the food and the tissue of our birds for this experiment, we used the multivitamin supplement as directed on our canary seed diet, providing ~10-15,000 IU retinol per kg diet. This is within the range measured by Wolf et al. (2000) to be needed for canary-typical circulating retinol levels in both WR and Y birds. These levels reported in Wolf et al. (2000)—roughly 0.7-0.9 µg retinol per mL plasma—are comparable to those reported in two other songbirds, domestic zebra finches (Blount et al. 2003) and wild great tits (Hörak et al. 2004). While the exact levels of retinol present in our birds are unknown, it is unlikely that our diet provided a level of retinol that is vastly different from the normal range for canaries and other songbirds.

The knockdown line retains traces of carotenoids in the body (line 25). Checking the concentration of carotenoids in the cited reference (Ref 10), the liver of white canaries contain about $2 \mu\text{g}\cdot\text{g}^{-1}$ of carotenoids instead of the typical $13 \mu\text{g}\cdot\text{g}^{-1}$ of yellow canaries (Toomey et al. 2017; please note that other coloured species can have much less carotenoids in the liver, e.g. house finches, McGraw et al. 2006). Since the liver is a fundamental organ for carotenoid metabolism it would be important to explain why these “traces” of carotenoids are negligible and their physiological action can be excluded. Would it be possible that low traces are enough for fulfilling antioxidants and/or immune enhancing functions and that surplus is used by yellow type to colour skin and feathers? I couldn't not find any data about carotenoid concentration in the blood, is this known?

RESPONSE: This is a very important question, and one we think is important for future studies to pursue. Ideally, we would have plasma carotenoid concentration measurements for all birds across multiple treatments, so that we could attempt to test for correlations between carotenoid levels and performance at an individual level, given that some WR canary tissues do contain measurable carotenoids. It is possible that the low levels of carotenoids in WR birds are sufficient for physiological benefit; it would be interesting to test whether there is some sort of “carotenoid threshold” below which we do start to see decreased performance, though this is unfortunately not possible with our dataset. The WR system is rich for future study, and we now briefly address this point in our discussion.

Importantly, our observation that the low levels of carotenoids in WR canary tissue were adequate for normal function presents a very serious challenge to conventional wisdom surrounding carotenoid limitation and costly allocation. If birds can meet their physiological needs with quantities of carotenoids that are insufficient to produce even a hint of yellow ornamentation, then a tradeoff between ornamentation and physiology is not a plausible explanation.

We did assess the carotenoid content (in collaboration with M. Toomey) of a small subset of canaries (4 Y and 4 WR). We opted not to include this data due to low sample size and high variation, and the patterns are the same as those presented in Toomey et al. (2017): $0.74 \pm 0.36 \mu\text{g}$ carotenoids per mL plasma for WR canaries, and 20.31 ± 21.26 for Y canaries. Note that McGraw et al. (2006) do not explicitly report total carotenoids, but we can estimate from Fig. 1 that wild, molting house finches possessed between ~8 (females) and ~9 $\mu\text{g}/\text{mL}$ total plasma carotenoids, and between ~15 (females) and ~25 $\mu\text{g}/\text{g}$ (males) total liver carotenoids. Our Y canaries therefore appear to have slightly lower liver and slightly higher plasma carotenoids than the molting house finches, while the WR birds possess a much lower concentration than either Y canaries or house finches in both tissues. We know of no other colorful bird species (and few dull bird species) that possess as low values for total carotenoid concentration as are found in WR canaries (e.g. Tella et al. 2004, supplementary table of mean plasma concentrations across 80 bird species). There is a reason that birds have SCARB1 to boost carotenoid uptake. Without this enzyme, passive diffusion of carotenoids provides very little material for physiological function.

The dose of carotenoids provided with the food was low, as stated by the authors in line 240. Would it be possible that providing them a greater amount of dietary carotenoids would have led to different results? I suggest addressing this issue in the discussion.

RESPONSE: We think it unlikely that a larger quantity of dietary carotenoids would have modified the results, given that our Y canaries had songbird-typical tissue carotenoid levels and possessed canary-typical coloration, reproduction, molt, and behavior. Indeed, providing birds with levels of carotenoid exceeding what they encounter in wild diets can actually begin to induce stress (Huggins et al. 2010, *Naturwissenschaften*). The carotenoid dose provided to canaries in this study enables lineages to produce color, sing, reproduce, and live for more than a decade in captivity. It was the correct dose to use in a comparison of the effects the knockdown of an uptake enzyme. See also our response to R3 below.

More information about the respiratory burst response (line 294-305) should be provided because this is not a very common technique used to assess the performance of the immune system. In particular, expected results in relation to the action of carotenoid should be clearly explained. Is it that carotenoids would increase the production of reactive oxygen species because they boost the immune system, thus the respiratory burst of heterophils? Please explain. Moreover, when discussing the results of the respiratory burst response it should be specified that they cannot be considered conclusive because of the very low sample size (6WR and 5 Y of different sexes, Line 311 and Table 1 extended data). Since it almost missed the significance this could be due to low power, thus further studies should be suggested to test if the higher respiratory burst observed in white canaries is really not significant.

RESPONSE: We believe that further explanation of the respiratory burst response is best presented in the Methods, as we are attempting to make the main text of the manuscript as concise as possible. We have added more explanation regarding the potential role(s) of carotenoids in this response to the Methods. The

reviewer has pointed out a key question within the carotenoid literature: while respiratory burst is often mentioned as an immune process in which carotenoids may potentially be involved, there is little understanding of whether this involvement would be as immune cell boosters or as antioxidants. While the carotenoid literature tends to suggest the latter, *in vitro* studies on mammalian cells indicate the former (reviewed briefly in Chew and Park, 2004, p. 295S). Because the specific mechanism for participation of carotenoids in oxidative burst has never been clearly articulated, we were skeptical about the hypothesis. We included measurements of oxidative burst because it is so often mentioned in the literature. The vagueness of the specific predictions related to oxidative burst are the failures of previous researchers who state the process as a key arena of carotenoid involvement. We were simply able to show that one measure of oxidative burst was not significantly affected by big differences in circulating carotenoids.

We have added a note regarding the small sample size of our respiratory burst tests in the caption to Figure 2.

Birds have been exposed to several treatments and several bleedings. A timetable of the study design (now there are some general indications about when a particular treatment has been done, but it is difficult to follow) and the amount of blood taken each time should be reported as repeated bleeding could interfere with several physiological pathways and individual condition.

RESPONSE: We agree that the timing of the various experimental procedures was confusing in-text. We have added Supplementary Fig. 1, which details the timeline of the three main experiments in which we manipulated and extracted blood from the birds (and lists the amount of blood taken at each point). We ensured that at least four weeks elapsed between segments for each bird so that no bird was bled more than ~150 μ L per month (the canaries tend to weigh 23-25g on average) both for animal welfare reasons and to minimize physiological effects of repeated bleeding.

Reviewer #2 (Remarks to the Author):

The authors use a novel system – the white strain of the domestic canary – to test the long held view that carotenoids enhance immune function or antioxidant defenses. The authors use a diversity of approaches to evaluate immune and antioxidant defenses in white canaries (with trace circulating & tissue carotenoids) and yellow canaries (with normal amounts of carotenoids). Their tests are comprehensive and they find no hint of a difference between the two strains. The experiment is elegant and timely – the authors should be commended for providing a robust and serious challenge to the widely touted assumption that carotenoids play important roles in key physiological functions. This is the kind of study that will cause many of us to go back to old published work and have a very serious think about alternative interpretations of the reported results.

The results of this study are novel, they will have undoubtedly a very high impact on the field, and they will be of wide interest to evolutionary ecologists and the wider field. The experiment is simple – which is its main strength. And the authors utilized fully four different approaches to test for differences between mutant and yellow strains – the study's other main strength. The statistics are fine and sample sizes are large enough to have detected any trends if there were any. There is no need for overly complex stats for this particular study.

I have no major concerns with the paper – the writing was clear and concise. One question that comes to mind, however, is why has it been the case that the hypothesis that carotenoids play a direct role in immune or antioxidant function has been such a pervasive feature of honest signalling theory? Is it just that we *wanted* it to be the case for facilitating a just-so story about signal costs? Or has any putatively positive evidence for the hypothesis simply been interpreted incorrectly? Perhaps this paper is not the platform for such discussions, but the question might be worth exploring in more detail.

RESPONSE: The senior author (G. Hill) can give his opinions on why the allocation tradeoff hypothesis become so entrenched over the past 15 years:

I think a few very prominent researchers in the field really liked the idea and they promoted it very hard. I am not claiming anything nefarious or unethical. It is really how science works. In the late 1990s, there was a general frustration at a lack of a coherent explanation for how carotenoids served as indicators of overall condition. When the resource allocation model was proposed, it seemed like a robust mechanism for honest signaling. Some leading thinkers in the field embraced and promoted the explanation. A popular idea gets tested a lot and gets hyped through lots of play in the literature, and researchers were inclined to see support for the popular hypothesis even in marginal data. Again, we all deal with biases and we all have pet ideas that we look to support. I think this is how Kuhn would say science inevitably works. And, the process of science works. No matter how popular an idea becomes, it can always be challenged with data.

James Dale

Reviewer #3 (Remarks to the Author):

The manuscript describes a study comparing the response of two breeds of domesticated canary to a variety of immune and oxidative challenges. The experimental challenges are start of the art, in particular the breadth they cover.

While the results are clear and conclusive, various aspects of the experimental design are relevant for its inference, which in my view is more restricted than what the authors imply.

Doubt that carotenoids per se are important antioxidants and immune stimulants dates back a decade. In the evolutionary ecology literature, the thinking has become more sophisticated after the initial proposal that the dual role of carotenoids as antioxidants and colourants directly enforces honesty on sexual signals. Various authors have pointed out that the role of carotenoids as antioxidants is possibly not large in birds (as also shown by references 6 and 7 in the manuscript). Indeed, the last 10 years of research have painted a more complex picture. Nonetheless, I agree with the authors that the field remains fairly muddled, despite some recent meta-analyses and reviews. Thus, it is critical that seminal contributions to high profile journals clarify the field, and make no unsubstantiated claims: in order to test for nutraceutical benefits, various realistic levels of supplementation against a homogenous, but natural, diet background of relevant study animals is critical. To test the proposed evolutionary ecological role of carotenoids in signalling, ecological and evolutionary relevance is important, that is natural conditions and carefully controlled life-history. In my view, this manuscript does not decisively contribute to either of these fields.

Yellow canaries are a domesticated breed, selected for high carotenoid incorporation into the plumage. Since plasma carotenoid levels and carotenoid processing machinery can also evolve rapidly in response to selection for ornamentation (Simons et al. 2014. Carotenoid-Dependent Signals and the Evolution of Plasma Carotenoid Levels in Birds. *Am Nat* 184:741–751) it is critical to know what other genetic and physiological changes have occurred in the canary breeds, that were both a result of independent artificial selection processes. For example, white canaries might have compensating mutations to deal with absence of carotenoids, yellow canaries might have been selected to be insensitive to physiological effects of dietary carotenoids.

RESPONSE: Carotenoid coloration, processing, and transportation can indeed evolve rapidly on an evolutionary time scale. Importantly, however, our WR and Y canaries differ essentially in only one gene—SCARB1—that inherits in a Mendelian recessive manner in our birds. The white canaries that we used are called “white recessives” because in crosses with yellow canaries, which occur frequently, the white phenotype behaviors as a recessive trait. There is no more chance for evolutionary mechanisms associated with the WR allele in aviary populations of canaries than there is for specific evolutionary mechanisms associated with coat color in Labrador Retrievers. It is one population of canaries with allelic variation in feather coloration. While we were unable to use multiple breeding seasons to create full-sibling WR and Y canary pairs for our main experiments, we did perform WR-Y crosses in several of our non-experimental birds, and the offspring are phenotypically indistinguishable from full-Y birds. Canary breeders regularly cross their genetic lines of color-bred canaries like these in order to maintain healthy, out-bred stock. While the exact origin of the WR canary is not as well-known as that of the red factor canary, there is no record of its existence prior to the early 1900s.

(Note that Toomey et al. 2017 compared white recessive, color-bred canaries to a wide range of canary breeds with yellow coloration but different body conformations and genetic histories, so there are likely even fewer genetic differences between our WR and Y birds of the same breed than were reported in that paper.)

While it is clear that both canary breeds respond similarly, although it seems the caveat is that this result depends on the diet containing sufficient retinol (vitamin A) – the specific deficiency that results in the white canaries as a result the mutation in SCARB1.

The white canaries critically depend on a dietary source of retinol (vitamin A). In natural situation, vitamin A levels are positively related to carotenoid plasma levels (Simons et al. 2014. An appraisal of how the vitamin A-redox hypothesis can maintain honesty of carotenoid-dependent signals. *Ecol Evol* 5:224–228), and this should be discussed as a potential experimental artifact or explanation for correlative patterns that are found in natural situations.

RESPONSE: We now address this important point in the Discussion.

Additionally, this is not a true experimental study, in order to conclude whether carotenoids boost immune or antioxidant defenses, both breeds should have been exposed to diets low and high in carotenoids and low and high in retinol. Non-experimental captive studies in general are not suitable to demonstrate allocation trade-offs and the evolutionary ecological relevance of the study is difficult to assess.

RESPONSE: Our goal was to test a critical assumption of the resource tradeoff hypothesis: that carotenoids play vital physiological roles in birds. We conducted what might be called an extreme test. We removed essentially all of the carotenoids in the bodies of a group of birds and subject them to environmental challenges. We compared the physiological responses of these carotenoid deficient birds to a group of birds that is permitted to circulate carotenoids. By any interpretation of the resource allocation hypothesis, the prediction is clear: loss of carotenoids should lead to loss of physiological function. There can be no tradeoff if carotenoids provide no critical physiological benefit. If we had found an effect then we could have proceeded to complicated experimental designs. But further manipulations are now meaningless. At a most extreme case of deprivation, loss of carotenoids induces no loss of function.

Here, we did not intend to demonstrate allocation tradeoffs (or a lack thereof). Indeed, as discussed in the following comment, finding a “natural” dietary carotenoid regimen at which to test for tradeoffs in the domestic canary may not be possible. While other experimental studies have used supplementation to create exaggerated carotenoid access in their research subjects, our system features exaggerated carotenoid restriction through the genetic knock-down inherent in the WR birds. Instead of testing for allocation tradeoffs, we instead test for a fundamental assumption in costly carotenoid tradeoff theory—that internal carotenoids are useful at all.

We offered a retinol dosage within the range recommended by Wolf et al. (2000) for WR canary welfare. Safely restricting retinol access was not feasible in birds so sensitive to deprivation. We absolutely agree that experimentally manipulating retinol access will be important for follow-up studies on other animal species.

The dietary composition is not very well justified, what can be considered a natural dietary carotenoid content for domestic yellow or white canaries? This is an important consideration, and I would have expected more of a discussion of this issue given the previous – and important - contributions by the authors on the importance of doing so (Koch et al. 2016. The Importance of Carotenoid Dose in Supplementation Studies with Songbirds. *PBZ* 89:61–71)

RESPONSE: We have added further detail about the justification for our canary diet in the Methods. In our study, the manipulation of carotenoid intake was built into the WR canaries themselves (irrespective of diet). At the same time, we did not want provide an over-abundance of dietary carotenoids to birds (like the “color feeding” often performed with colored canaries) because exceptionally large carotenoid intake could itself be problematic for the Y birds (Huggins et al. 2010). The diet we selected—a plain canary seed mix with a carotenoid-free multivitamin supplement (containing retinol)—is very typical of the diets provided to domestic canaries by aviculturists (see Methods). On this diet, canaries in aviaries around the world produce yellow coloration and full song, reproduce, and live for a decade or more.

Importantly, we observed most of the birds through two molting and breeding seasons (a subset of non-experimental birds were allowed to breed) and noted no aberrant behaviors. Necropsies on birds that died over the course of the long-term colony’s maintenance revealed no signs of retinol deprivation or other dietary-related pathologies.

No information on breeding and rearing conditions of canaries, this is important given the long-lasting effects of such conditions (Evans SR, Sheldon BC. 2012. Quantitative Genetics of a Carotenoid-Based Color: Heritability and Persistent Natal Environmental Effects in the Great Tit. *Am Nat* 179:79–94.)

RESPONSE: Birds in this study were raised off-site by canary aviculturists, all under the same conditions in large, long-term breeding flocks fed the same diets. Certainly, there was no differential treatment of Y and WR birds. We expect minimal variation in the conditions experienced by different birds during their early lives. We now state this in the methods.

Reviewers' comments:

Reviewer #1 (Remarks to the Author):

I have read with attention the rebuttal of the authors and I think they addressed very well all my comments. My only further suggestion is to openly state in the discussion that the description of the redox state has some limits, as any biomarker of oxidative damage has been used in the study.

Reviewer #3 (Remarks to the Author):

The authors provide important additional information in their revised manuscript and in the cover letter, that gives greater confidence in the results.

Given the central importance of the fact that the two canaries are genetically identical apart from a single point mutation, it is warranted to reiterate this in the methods. This was not directly obvious to a non-initiate, also not after reading the referenced paper.

Finally, I feel the authors still overstate the broad view of the literature, which in my view detracts from the manuscript.

l. 120-121: regarding the evolutionary literature, while some uncritical assertions have certainly been made, overall, it is critical of the idea that carotenoids directly boost immune and antioxidant performance. This has been broadly regarded as a plausible (and attractive, I warrant) hypothesis, for which some support certainly does exist, but also plenty of nuanced discussion and critique, and some attempts at synthesis across disparate results (e.g. Olson VA, Owens IPF. 1998. Costly sexual signals: are carotenoids rare, risky or required? *Trends Ecol Evol* 13:510–514;

Costantini D, Møller AP. 2008. Carotenoids are minor antioxidants for birds. *J Anim Ecol* 22:367–370; Pérez-Rodríguez L. 2009. Carotenoids in evolutionary ecology: re-evaluating the antioxidant role. *Bioessays* 31:1116–1126; Bertrand et al. 2006. Do carotenoid-based sexual traits signal the availability of non-pigmentary antioxidants? *J Exp Biol* 209:4414–4419; Simons et al. 2014; Carotenoid-Dependent Signals and the Evolution of Plasma Carotenoid Levels in Birds. *Am Nat* 184:741–751. Simons. 2012. What Does Carotenoid-Dependent Coloration Tell? Plasma Carotenoid Level Signals Immunocompetence and Oxidative Stress State in Birds—A Meta-Analysis. *PLoS ONE* 7:e43088–14)

For instance, ref 9, cited at l. 121 in support of a 'widely espoused assumption' is a thorough and balanced review of the literature, and concludes "Studies investigating the physiological trade-offs between ornamental and physiological uses of carotenoids have yielded inconsistent results (Svensson and Wong 2011)". Svensson and Wong also note that "Current controversies may be resolved through a more careful regard of this complexity, and of the immense taxonomic variability of carotenoid metabolism.". Following on from this suggestion seems more productive than setting up a strawman, as is now the case at l. 120-121. It undermines the credibility of this ms.

I suggest re-word l. 23: "These results add further challenge to the assumption that carotenoids...".

Also, regarding the nutraceutical literature, this mostly concerns mammals (mice, humans), as the senior author pointed out (Hill GE. 1999. Is There an Immunological Cost to Carotenoid-Based Ornamental Coloration? *Am Nat* 154:589–595.), the role for carotenoids in mammals is possibly different than in birds, unless further research since this 1999 paper has altered this view? If not, it feels disingenuous to extrapolate from this study to vertebrates and some finetuning of statements would improve the impact of the ms, eg. L. 26-27, add 'in birds'? or, 'in captive birds'?

l. 120: note spelling: nutraceutical not nutraseutical.

The manuscript is lacking a broader, more constructive discussion for example, to propose hypotheses that can reconcile these results in the mutant canaries with the body of evidence for positive correlations between carotenoid and aspects of health and experimental studies supporting these.

Reviewer #1 (Remarks to the Author):

I have read with attention the rebuttal of the authors and I think they addressed very well all my comments. My only further suggestion is to openly state in the discussion that the description of the redox state has some limits, as any biomarker of oxidative damage has been used in the study.

RESPONSE: We have added specific mention of this limitation to the discussion, as suggested.

Reviewer #3 (Remarks to the Author):

The authors provide important additional information in their revised manuscript and in the cover letter, that gives greater confidence in the results.

Given the central importance of the fact that the two canaries are genetically identical apart from a single point mutation, it is warranted to reiterate this in the methods. This was not directly obvious to a non-initiate, also not after reading the referenced paper.

RESPONSE: Thank you for the suggestion; the relationship between the WR and Y birds is, of course, fundamental to the premise of our paper. As such, we have considerably elaborated on the nature of our birds in the first paragraph of the Methods section.

Finally, I feel the authors still overstate the broad view of the literature, which in my view detracts from the manuscript.

RESPONSE: This is an important issue that we hope to avoid in our manuscript, and we appreciate the reviewer providing his/her perspective here. We have tried to change wording in our manuscript to better frame the results, without diminishing the impact of the study. See responses below.

l. 120-121: regarding the evolutionary literature, while some uncritical assertions have certainly been made, overall, it is critical of the idea that carotenoids directly boost immune and antioxidant performance. This has been broadly regarded as a plausible (and attractive, I warrant) hypothesis, for which some support certainly does exist, but also plenty of nuanced discussion and critique, and some attempts at synthesis across disparate results (e.g. Olson VA, Owens IPF. 1998. Costly sexual signals: are carotenoids rare, risky or required? *Trends Ecol Evol* 13:510–514; Costantini D, Møller AP. 2008. Carotenoids are minor antioxidants for birds. *J Anim Ecol* 22:367–370; Pérez-Rodríguez L. 2009. Carotenoids in evolutionary ecology: re-evaluating the antioxidant role. *Bioessays* 31:1116–1126; Bertrand et al. 2006. Do carotenoid-based sexual traits signal the availability of non-pigmentary antioxidants? *J Exp Biol* 209:4414–4419; Simons et al. 2014; Carotenoid-Dependent Signals and the Evolution of Plasma Carotenoid Levels in Birds. *Am Nat* 184:741–751. Simons. 2012. What Does Carotenoid-Dependent Coloration Tell? Plasma Carotenoid Level Signals Immunocompetence and Oxidative Stress State in Birds—A Meta-Analysis. *PLoS ONE* 7:e43088–14)

For instance, ref 9, cited at l. 121 in support of a ‘widely espoused assumption’ is a thorough and balanced review of the literature, and concludes “Studies investigating the physiological trade-offs between ornamental and physiological uses of carotenoids have yielded inconsistent results (Svensson and Wong 2011).”. Svensson and Wong also note that “Current controversies may be resolved through a more careful regard of this complexity, and of the immense taxonomic variability of carotenoid metabolism.”. Following on from this suggestion seems more productive than setting up a strawman, as is now the case at l. 120-121. It undermines the credibility of this ms.

RESPONSE: One of the challenges we faced in preparing this manuscript is how to articulate that the benefits of carotenoids have indeed been challenged and debated previously in a wide variety of papers, including the excellent citations listed here. However, there still seems to be a pervasive fixation on testing for or even assuming the benefits of carotenoids across carotenoid signaling literature. Often, it seems as though studies that do not find expected relationships between carotenoids and internal quality attribute their results to a failure to test the right parameters or other experimental issues. There is the general expectation that even though carotenoid resource tradeoffs may or may not underlie signal honesty, internal carotenoid pigments SHOULD be beneficial, at the very least because they often behave as antioxidants *in vitro*. Indeed, reconciling the *in vitro* behavior of carotenoids with their effects in a live animal body is a central challenge in the carotenoid literature.

Anyway, while an in-depth discussion of the debate surrounding carotenoid benefits and honest signaling is beyond the scope of this short manuscript, we have aimed to reword key phrases to try to better frame our study within the broader carotenoid literature (including lines 30-31 and lines 120-121—we removed the

phrase in question entirely from the latter). While we continue to pursue a relatively concise manuscript, we hope that we now avoid over-simplifying the carotenoid literature or over-interpreting our results.

I suggest re-word I. 23: "These results add further challenge to the assumption that carotenoids...".

RESPONSE: We have made the suggest change to the wording in this sentence.

Also, regarding the nutraceutical literature, this mostly concerns mammals (mice, humans), as the senior author pointed out (Hill GE. 1999. Is There an Immunological Cost to Carotenoid-Based Ornamental Coloration? Am Nat 154:589–595.), the role for carotenoids in mammals is possibly different than in birds, unless further research since this 1999 paper has altered this view? If not, it feels disingenuous to extrapolate from this study to vertebrates and some finetuning of statements would improve the impact of the ms, eg. L. 26-27, add 'in birds'? or, 'in captive birds'?

RESPONSE: We have added the qualifer "in birds" as suggested, since we agree that the processes described here may be taxon-specific. We have retained a reference to the fact that carotenoids are also often considered important antioxidants in the "nutraceutical" (i.e. mammalian) literature because it demonstrates how widespread the claims of carotenoid benefits are. Assessing SCARB1 knock-out mice would be an important step toward testing the function of carotenoids in mammals.

I. 120: note spelling: nutraceutical not nutraceutical.

RESPONSE: We removed the phrase containing this error.

The manuscript is lacking a broader, more constructive discussion for example, to propose hypotheses that can reconcile these results in the mutant canaries with the body of evidence for positive correlations between carotenoid and aspects of health and experimental studies supporting these.

RESPONSE: In our original submission, we intended to keep discussion of alternative hypotheses to a minimum both for the sake of staying concise and to avoid speculation on ideas that were not tested here. While we still believe that our manuscript is best kept short, we have added additional information on alternative hypotheses that do not require carotenoids to provide direct physiological benefits. Finding ways to clearly test these alternatives empirically is a critical next step to advancing the carotenoid signaling literature.